# Regional, institutional, and departmental factors associated with gender diversity among BS-level chemical and electrical engineering graduates

**Laura R. Jarboe** *

Department of Chemical and Biological Engineering, Iowa State University, Ames, Iowa, United States of America

* ljarboe@iastate.edu

**Data Availability Statement:** All relevant data are within the manuscript and its Supporting Information files.

## Abstract

Engineering remains the least gender diverse of the science, technology, engineering and mathematics fields. Chemical engineering (ChE) and electrical engineering (EE) are exemplars of relatively high and low gender diversity, respectively. Here, we investigate departmental, institutional, and regional factors associated with gender diversity among BS graduates within the US, 2010–2016. For both fields, gender diversity was significantly higher at private institutions ($p < 1 \times 10^{-6}$) and at historically black institutions ($p < 1 \times 10^{-5}$). No significant association was observed with gender diversity among tenure-track faculty, PhD-granting status, and variations in departmental name beyond the standard "chemical engineering" or "electrical engineering". Gender diversity among EE graduates was significantly decreased ($p = 8 \times 10^{-5}$) when a distinct degree in computer engineering was available; no such association was observed between ChE gender diversity and the presence of biology-associated degrees. States with a highly gender diverse ChE workforce had a significantly higher degree of gender diversity among BS graduates ($p = 3 \times 10^{-5}$), but a significant association was not observed for EE. State variation in funding of support services for K-12 pupils significantly impacted gender diversity of graduates in both fields ($p < 1 \times 10^{-3}$), particularly in regards to instructional staff support ($p < 5 \times 10^{-4}$). Nationwide, gender diversity could not be concluded to be either significantly increasing or significantly decreasing for either field.

## Introduction

The collective intelligence of a group has been found to correlate not with the average of maximum intelligence of individual group members, but with the group's proportion of females [1, 2]. The proportion of females within a group has also been reported to increase the frequency of cooperative interactions [3]. However, women accounted for less than 30% of the science, technology, engineering and mathematics (STEM) American labor force in 2015 [4].

In 2018, a female chemical engineer was awarded the Nobel Prize in Chemistry, and two female engineers were elected to the US House of Representatives. However, engineering

**Funding:** The author(s) received no specific funding for this work.

**Competing interests:** The authors have declared that no competing interests exist.

remains the least gender diverse STEM discipline, with only 15% of the engineering workforce being female, in contrast to 60% of social scientists being female [4]. The gender skew of engineering relative to other STEM fields is apparent in a survey of US students enrolled in introductory college-level English. Specifically, being female was positively associated with selection of a science career but negatively associated with selection of an engineering career [5]. Engineering is also distinct from other types of STEM in the persistence of females in academic positions: female tenure-track engineering faculty have a significantly lower probability of retention over a 15 year period relative to their male colleagues, while no significant difference was observed between genders for physical and mathematical sciences, biological and biomedical sciences, or agriculture and natural resources [6].

Within the various engineering fields, electrical engineering (EE) and chemical engineering (ChE) are hallmarks of relatively low and relatively high gender diversity, respectively. The problem of low gender diversity in EE has been the topic of discussion for more than 25 years [7] and EE was used in 1996 as a representative male-skewed field during characterization of gender stereotypes [8]. A variety of explanations have been proposed for the relatively high proportion of females in chemical engineering, including its reputation as the most challenging engineering program and its similarity to cooking [9].

Middle and high school students displayed a gender-dependent disparity in interest in both ChE and EE, with males showing a 4-fold higher interest in electrical engineering and 2-fold higher interest in chemical engineering relative to females [10]. A separate assessment of interest in EE reported a similar gender skew in high school students but not in elementary school students [11]. Multiple studies have investigated external factors that influence the decision of high school students to pursue engineering as a career and that are associated with the retention of undergraduates through degree completion. For example, female high school students with a higher proportion of female math and science teachers were more likely to complete a BS in physical science, engineering or mathematics [12]. At the university level, learning environments that emphasize respect for students and regular interaction with faculty were identified as especially impactful for female engineering students [13–15]. Matching of female engineering students with a female peer mentor and placing in small groups with female peers during their first year of undergraduate studies has also been shown to be beneficial [16, 17].

Internal factors also influence the decision of field of study and persistence in that field through graduation and career establishment. It has been proposed that underrepresentation of females in certain STEM fields continues because careers in these fields are perceived as having a lower possible contribution to fulfilling communal goals, where fulfilling communal goals is high priority for female students [18, 19]. It has also been shown that students' perception of gender bias is a substantial driver of the gender disparity across college majors [20] and that deviation from gender stereotypes is judged more strongly for females than for males [8].

Here, we quantify the gender diversity among recent BS graduates of electrical engineering and chemical engineering among a focal pool of 95 institutions across the US. This diversity data is compared to a variety of department-, institution-, state- and region-specific properties, with the goal of identifying factors and strategies that enable and support diversity at the level of gender, and possibly increased participation from members of other underrepresented groups.

## Materials and methods

### Data sources

All of the data used in the analysis is presented in the accompanying supplemental tables.

Numbers of total BS graduates and female BS graduates for individual institutions were obtained from the Integrated Postsecondary Education Data System (IPEDS) using ChE (CIP code 14.07) or EE (CIP code 14.10) as first or second major. Nationwide data on BS graduate numbers and overall gender diversity within fields was obtained from the American Society for Engineering Education (ASEE)'s annual "Engineering by the Numbers" reports [21–27] and Appendix Table 2–21 of the 2018 the National Science Foundation (NSF) "Science & Engineering Indicators" report [4]. Data regarding the existence of unique ChE and EE departments, departmental name, and the number of other available engineering majors was obtained from the Accreditation Board for Engineering and Technology (ABET) using the data file last updated on Oct 1, 2017. Number and gender distribution of tenure-track faculty were obtained from ASEE college profiles for 2005, 2010 and 2015, and an average value was calculated across these three years. Average employment data for 2006–2010 was obtained from the United States Census Bureau American Community Survey using occupation code 1350 (SOC 17–2041) for ChE and occupation code 1410 (SOC 17–2070) for EE. Data for state spending on K-12 education was obtained from the annual "Public Education Finances" report published by the US Census Bureau. Values were obtained for all years between academic year 1996/97 and academic year 2011/12, and an average value calculated for each state. State-specific values of the gender earnings ratio were obtained from the American Association of University Women (AAUW) annual report [28].

### Statistical methods

Significance was assessed using regression and one-way ANOVA tools in Microsoft Excel with a confidence level of 99.9%. To reduce the probability of Type I errors, a p-value of 0.001 was applied as the criterion for significance. Outliers were identified according to Tukey's method (k = 1.5).

Box plots were generated using BoxPlotR (http://shiny.chemgrid.org/boxplotr/). Center lines show the medians within each category, box edges indicate the 25th and 75th percentiles, and box width is proportional to the square root of the number of observations.

## Results

### The dataset

As of October 1, 2017 ABET lists more than 300 unique undergraduate EE programs in the US and more than 160 ChE programs. Approximately 160 schools are accredited for both majors. From this list, 95 focal schools were selected. We aimed to include at least one, or preferably two, institutions per state. Puerto Rico and the District of Columbia are represented in this pool, while Alaska, Hawaii and Vermont are not, due to the lack of any institutions with both an EE and ChE ABET-accredited program. Schools were also selected based on ASEE reporting of the largest ChE and EE programs [22–27], and to include Historically Black Colleges and Universities (HBCUs) and undergraduate-focused institutions. These 95 focal institutions consist of 21 private institutions and five HBCUs (Table 1, S1 Table). Three of the HBCUs are private, two are public. Seventeen institutions do not offer a PhD in either program. Three offer a PhD in ChE but not EE, and six offer an EE PhD but not ChE.

Graduation data for students with EE or ChE as first or second major was obtained from IPEDS for 2010–2016 (Table 1, S1 Table). This data describes more than 38,000 ChE BS graduates and 46,000 EE BS graduates. Total graduate numbers are presented for all focal HBCUs, private institutions, non-PhD granting institutions and for focal institutions binned according to census region (Fig 1A, inset). Of the ChE BS graduates characterized here, 16.9% graduated from a private institution, 1.6% graduated from an HBCU, and 9.4% graduated from a non-

**Table 1. Overview of focal institutions and BS graduate data, 2010–2016.**

| | Chemical Engineering (ChE) | Electrical Engineering (EE) |
|---|---|---|
| Number of total focal institutions | 95 | |
| Number of private institutions | 21 | |
| Number of HBCU | 5 | |
| Number of PhD-granting institutions | 72 | 74 |
| Total number of BS graduates across all focal institutions | 38,620 | 46.222 |
| Range of total BS graduates per institution (average) | 32–1,054 (407) | 15–2,308 (487) |
| Percent of total BS graduates across all focal institutions as female | 33.0 | 12.6 |
| Range of percent of total BS graduates as female per institution (average) | 15–64 (34) | 5–34 (13) |

PhD-granting department. These values are 14.0, 1.5 and 10.5% for EE BS graduates. Across all 95 focal institutions during this time period, ChE programs averaged 407 total BS graduates, with per institution values ranging from 32 to 1,054 BS graduates. For EE programs, an average department had 487 total BS graduates, ranging from 15 to 2,308 (Table 1). Outliers in terms of the high number of ChE or EE BS graduates are marked in (Fig 1A), with two institutions being outliers for both ChE and EE. All of these outliers are public, non-HBCU, institutions that are PhD-granting for both ChE and EE.

ASEE reporting quantified 54,451 ChE graduates and 74,788 EE graduates between 2010 and 2016 (S1 Table). Thus, the 95 focal institutions selected here account for 70.9% and 61.8% of the ChE and EE graduates, respectively, reported by ASEE. Records from NSF quantify 52,031 ChE BS graduates and 109,915 EE BS graduates between 2010 and 2015 (S1 Table). Note that 2016 NSF data was not available at the time of preparation of this manuscript.

Among the ChE and EE BS graduates at the 95 focal institutions, 33.0% and 12.6% were female, respectively (Table 1). In comparison, the bulk averages for the ASEE data are 33.4 and 13.5%, respectively (2010–2016) and 30.4 and 11.2% for NSF reporting (2010–2015). The fact that gender diversity is substantially lower for both fields in the NSF dataset relative to ASEE tracking suggests that institutions not participating in ASEE reporting have lower gender diversity than those that do self-report to ASEE.

The bulk gender diversity values described above for the 95 focal institutions selected here are generally consistent with existing reports by ASEE and NSF. However, these bulk values do not convey the degree of variability between institutions and regions. Among the 95 selected focal schools, the percent of female ChE BS graduates from individual institutions ranged from 15 to 64% (Table 1, Fig 1B). For EE graduates, values ranged from 5 to 34%. Three institutions, all of which are private institutions, were statistical outliers in terms of their high gender diversity among both ChE and EE graduates. Two of these are non-HBCU institutions that grant PhDs in both ChE and EE, and one is an HBCU that does not grant PhDs in either field.

Comparison of the total number of BS graduates, regardless of gender, on a per-institution basis reveals a significant correlation between the number of ChE and EE graduates (Table 2). On average, EE departments produce 30±10% more graduates than ChE departments at the same institution. This trend is conserved when institutions are binned as HBCUs, private, non-PhD granting or according to US census division (Fig 1A, inset). However, it should be noted that within US census region 9 (Pacific), EE departments produced an average of 86% more BS graduates than ChE departments. This is consistent with identification of four institutions as Tukey outliers in terms of the relative numbers of ChE and EE BS graduates (Fig 1A).

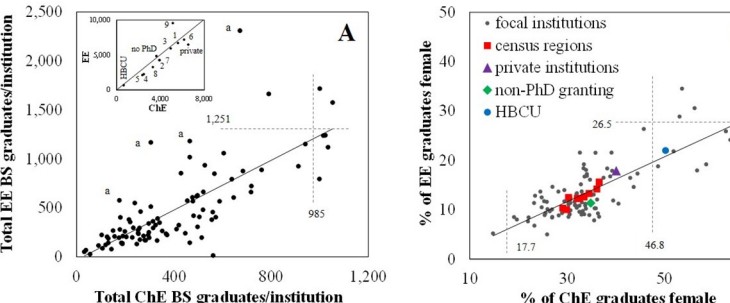

**Fig 1. Broad view of total numbers, gender diversity, variability by census region and institution type.** (A) EE departments produce, on average, 30±10% more BS graduates than ChE departments. The dashed lines indicate Tukey's fences. The inset shows the 9 US census regions and binned data for all focal private, HBCU and non PhD-granting institutions with the trendline from the all-institutions analysis shown for comparison. US Census Regions are: 1, New England; 2, Middle Atlantic; 3, East North Central; 4, West North Central; 5, South Atlantic; 6, East South Central; 7, West South Central; 8, Mountain; 9, Pacific. [a]Tukey outliers in terms of the EE/ChE ratio. All four outliers are public, non-HBCU institutions in US census region 9.(B) On average, ChE departments have 2.2±0.2-fold higher gender diversity among BS graduates than EE departments. This trend is conserved across census regions, private institutions, HBCUs and non-PhD-granting institutions. The dashed lines indicate Tukey's fences.

These four schools are all public non-HBCU institutions in California and Washington and produce more than 2.5 EE graduates per each ChE graduate.

When a similar analysis was applied to gender diversity among these graduates, it was observed that BS graduates from ChE departments are significantly more gender diverse than

**Table 2. Statistical analysis of institutional, college, and departmental factors.** A p-value less than 0.001 was considered statistically significant, with values meeting this criterion shown in bold. Slope values and the associated standard deviation are given as the change in the second variable (y) relative to changes in the first variable (x), with entries listed as x vs y. Slope values are provided only for relationships that met the significance criterion.

|  | ChE | EE |
|---|---|---|
| **Institutional Level** | | |
| Total number of BS graduates: ChE vs EE | $R^2 = 0.58$, **p = 4x10$^{-19}$**, slope = 1.3 ±0.1 | |
| % of BS graduates female: ChE vs EE | $R^2 = 0.58$, **p = 3x10$^{-19}$**, slope = 0.45 ±0.04 | |
| Institution type (public, private, HBCU, non-HBCU) vs % of BS graduates female | ANOVA **p = 5x10$^{-13}$** | ANOVA **p = 8x10$^{-15}$** |
| **College and Departmental Level** | | |
| % of tenure-track faculty female: ChE vs EE | $R^2 = 0.07$, p = 0.01 | |
| % TT faculty female vs % BS graduates female | $R^2 = 0.001$, p = 0.7 | $R^2 = 0.001$, p = 0.7 |
| Average # of TT faculty/total BS graduates 2010–2016 vs % BS graduates female | $R^2 = 0.01$, p = 0.2 | $R^2 = 0.11$, p = 0.001 |
| PhD-granting vs non-PhD granting | t-test: p = 0.01 | t-test: p = 0.9 |
| Variation in departmental name | ANOVA p = 0.97 | ANOVA p = 0.02 |
| Availability of distinct degree in competing engineering program | Bio* t-test: p = 0.02 | Computer t-test: **p = 8x10$^{-5}$** |
| Institution type (public, private, HBCU, non-HBCU) vs # of other engineering degrees available | ANOVA **p = 0.0007** | |
| # of other engineering degrees available vs % BS graduates female | $R^2 = 0.07$, p = 0.01 | $R^2 = 0.11$, **p = 0.00099**, slope = -0.6±0.2 |

EE departments (Fig 1B, Table 2). On average, the percent of female graduates was 2.2 ±0.2-fold higher for ChE departments than EE departments at the same institution and this trend was conserved when institutions were binned according to type and census region (Fig 1B). No institutions were observed as statistical outliers in terms of the relative gender diversity for EE relative to ChE graduates.

The adherence to this relationship between ChE and EE gender diversity across institution types and census regions indicates that there are regional- or institution-specific factors at play, independent of the factors specific to these departments at a single institution.

## Variation according to institution type

Consistent with previous reports [15], the distinction between HBCU and non-HBCU was significantly associated with gender diversity in both ChE and EE (Fig 2, Table 2). A significant association was also observed for the public vs private distinction for both ChE and EE (Table 2). These results indicate that HBCUs and private institutions have properties and/or employ strategies that support gender diversity across both types of engineering. The high persistence of female ChE students at HBCUs has previously been attributed to small college size and a climate that allows students to form close relationships with faculty [15].

## College- and departmental-specific factors

A variety of studies have concluded that gender diversity at the faculty level is associated with undergraduate retention and graduation within STEM [29, 30]. Data regarding tenure track (TT) faculty in 2005, 2010 and 2015 was obtained from the ASEE database for 93 of the 95 focal institutions (Fig 3A). Two of the focal institutions had no female TT faculty in either EE or ChE in all three sampling years. One institution, a public, non-PhD-granting, non-HBCU, was a Tukey outlier in terms of the high degree of gender diversity among TT faculty in both fields.

Unlike the significant correlation of institution-specific gender diversity values for ChE and EE BS graduates, the institution-specific gender diversity values for TT ChE and TT EE faculty were not significantly associated (Table 2). Surprisingly, no significant correlation was observed between TT faculty gender diversity and BS graduate gender diversity for either ChE or EE (Table 2, Fig 3B). The ratios of TT faculty/BS graduate were analyzed similarly, also with no observation of significant trends for either major (Table 2). Our results may differ from previous reports in that here we have only tracked the reported distribution of TT faculty. Actual teaching loads and the gender distribution of non-tenure track faculty have not been accounted for in this analysis.

Previous reports have found that female students and minority students have increased persistence in STEM fields at institutions that do not have a graduate program [31]. Binning of our 95 focal schools according to the presence or absence of a PhD program did not qualify as statistically significant for either ChE or EE in this study (Table 2).

The relatively high gender diversity in Chemical Engineering is often attributed to its association with biological applications [9]. Nearly half (45%) of ChE departments in our focal pool include some form of "bio" in the department name (S1 Table), while others contain "materials" (8%), "environmental" (3%) or other terms (6%), but these variations did not meet the criterion for significant association with gender diversity (Table 2). Also, while 64% of focal schools are accredited to offer a degree in bioengineering, biomedical engineering or biological engineering, the availability of the separate degree was not significantly associated with gender diversity among ChE BS graduates (Table 2).

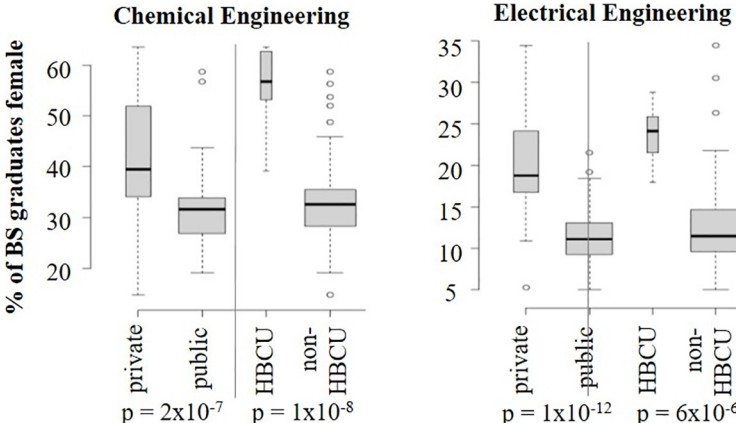

**Fig 2. Institutional characteristics.** Institution type significantly impacts gender diversity for both ChE and EE in terms of public vs private and HBCU vs non-HBCU.

Electrical Engineering is sometimes combined with Computer Engineering and/or Computer Science. Nearly three-fourths (73%) of the focal EE departments include "Computer" in the department name, 3% include "systems" and 3% include other terms (S1 Table), but as with ChE, variations in the departmental name did not meet the criterion for significant association with gender diversity (Table 2). Among the 95 focal schools, 85% have ABET accreditation to also offer a degree in Computer Engineering, and 67% in Computer Science. However, schools that do not offer a Computer Engineering degree have significantly higher gender diversity among EE BS graduates (Table 2) relative to those that do offer a Computer Engineering degree (Fig 3C).

Within the US, accreditation is available for more than 30 types of engineering degrees. According to the 2017 ABET records, the 95 focal schools in this study ranged from offering one other type of engineering degree beyond ChE and EE to offering 16 other types of engineering degrees (S1 Table). On average, institutions offered 7.4 other types of engineering degrees, with Mechanical Engineering and Civil Engineering being the two most common other majors, offered at 98.9% and 91.6% of focal institutions. The number of other degrees offered was found to vary significantly according to institution type (Table 2). On average, private institutions offered 5.4 additional degrees, public institutions offered 7.9, HBCUs offered 4.2 and non-HBCUs offered 7.5. The number of other degrees offered beyond ChE and EE

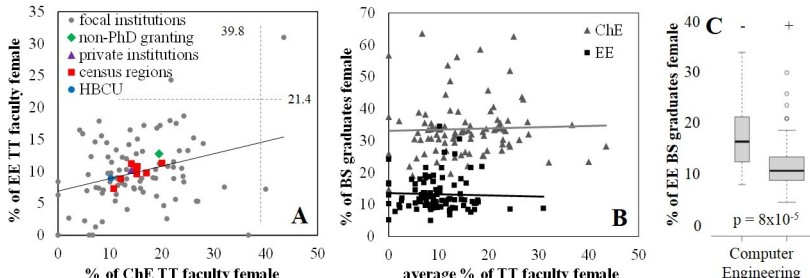

**Fig 3. Departmental characteristics.** (A) Institution-specific gender diversity values for tenure-track (TT) faculty. Dashed lines indicate Tukey's fences. (B) The gender composition of TT faculty is not significantly associated with gender diversity among BS graduates in either field. (C) Institutions that are ABET accredited for a distinct degree in Computer Engineering have significantly lower gender diversity among EE graduates.

was found to be significantly, and negatively, associated with gender diversity among EE BS graduates, but not ChE (Table 2).

While these results provide some insight into factors significantly associated with gender diversity among EE graduates, no college- or departmental-level factors were identified that were significantly associated with gender diversity among ChE graduates.

## Regional variation: Overview

In order to investigate regional factors associated with gender diversity among BS graduates, data for the 95 focal schools was binned according to state. While no significant differences were observed between states or census regions (Table 3, S2 Table), the significant correlation of gender diversity between ChE and EE BS graduates that was observed at the institution level (Fig 1B) was conserved at the state level (Fig 4A). On average, the pool of ChE BS graduates in each state was 2.9±0.3-fold more gender diverse than the pool of EE BS graduates in the same state (Table 3). Puerto Rico and the District of Colombia were Tukey outliers in terms of the high gender diversity among BS graduates in both fields, while Massachusetts was an outlier for EE but not ChE.

## Regional variation: Employment trends

Employment data was obtained for 2006–2010 for 46 states and the District of Columbia (S3 Table). South Dakota was excluded from this analysis due to the reporting of a very small number of practicing ChE and, as above, Alaska, Hawaii and Vermont are not represented in our focal pool. Census data was not available for Puerto Rico.

One possible motivation for students choosing to pursue a career in chemical engineering or electrical engineering could be exposure to full-time chemical engineers and electrical engineers. In all states, chemical engineers and electrical engineers, regardless of gender, accounted for less than 0.4% of full-time workers (Fig 4B). Texas, Louisiana and Delaware are Tukey outliers in terms of the high prevalence of full-time chemical engineers, while New Hampshire is an outlier in terms of the high prevalence of full-time electrical engineers. However, there is no significant association between the prevalence of chemical engineering and electrical engineering as full-time jobs and the gender diversity among recent BS graduates in these fields (Table 3).

It seems possible that exposure of female students to female working engineers could promote selection of ChE and EE as fields of study. US census data not only quantifies the number of full-time ChE and EE in each state, but also quantifies how many females were in each sample group (S3 Table). The reported percent of practicing ChE in each state who were female was as high as 43% (New Mexico), with an average of 14%. For EE, this metric was as high as 12% (New Jersey), with an average of 7.2%. This gender diversity among working engineers and among recent BS graduates was significantly associated for ChE but not EE (Fig 4C, Table 3). No significant association was observed between the gender diversity of new BS graduates and either: the percent of all full-time female workers who were a ChE or EE; or the percent of all full-time workers who were a female ChE or EE (Table 3).

Women working full-time in STEM have a lower median annual salary than men, even among new graduates with minimal differences in training and experience [4, 32]. The 2016 state-specific earnings ratio reported by AAUW ranged from 0.70 to 0.89 [28]. However, no significant association was observed between this metric and gender diversity among new BS graduates in the corresponding state (Table 3).

These results show that states that have higher gender diversity among practicing chemical engineers also have higher gender diversity among ChE BS graduates. While this relationship

**Table 3. Statistical analysis of state-specific factors.** A p-value less than 0.001 was considered statistically significant, with values meeting this criterion shown in bold. Slope values and the associated standard deviation are given as the change in the second variable (y) relative to changes in the first variable (x), with entries listed as x vs y. Slope values are provided only for relationships that met the significance criterion.

| | ChE | EE |
|---|---|---|
| Census region vs % of BS graduates female | ANOVA p = 0.1 | ANOVA p = 0.2 |
| State vs % of BS graduates female | ANOVA p = 0.1 | ANOVA p = 0.8 |
| % BS graduates female: ChE vs EE | $R^2 = 0.60$, **$p = 8 \times 10^{-11}$**, slope = 0.34±0.04 | |
| % of full-time jobs ChE vs EE | $R^2 = 0.01$, p = 0.4 | |
| % of full-time jobs ChE or EE vs % of BS graduates female | $R^2 = 0.01$, p = 0.4 | $R^2 = 0.00$, p = 0.7 |
| % of full-time ChE or EE workers who are female vs % of BS graduates who are female | $R^2 = 0.33$, **$p = 3 \times 10^{-5}$**, slope = 0.5±0.1 | $R^2 = 0.15$, p = 0.007 |
| % of full-time female workers who are ChE or EE vs % of BS graduates who are female | $R^2 = 0.00$, p = 0.8 | $R^2 = 0.06$, p = 0.09 |
| % of all full-time workers who are female ChE or female EE vs % of BS graduates who are female | $R^2 = 0.00$, p = 0.8 | $R^2 = 0.07$, p = 0.07 |
| 2016 earnings ratio vs % of BS graduates female | $R^2 = 0.05$, p = 0.1 | $R^2 = 0.06$, p = 0.09 |
| **Expenditure ($) per K-12 pupil vs % of BS graduates female** | | |
| Total | $R^2 = 0.13$, p = 0.01 | $R^2 = 0.14$, p = 0.008 |
| Instruction | $R^2 = 0.05$, p = 0.1 | $R^2 = 0.07$, p = 0.07 |
| "Other" (not instruction or support services) | $R^2 = 0.01$, p = 0.6 | $R^2 = 0.01$, p = 0.6 |
| All support services | $R^2 = 0.28$, **$p = 1 \times 10^{-4}$**, slope = 0.004±0.001 | $R^2 = 0.23$, **$p = 6 \times 10^{-4}$**, slope = 0.0019±0.0005 |
| Pupil support services | $R^2 = 0.19$, p = 0.002 | $R^2 = 0.11$, p = 0.02 |
| Instructional staff support services | $R^2 = 0.38$, **$p = 3 \times 10^{-6}$**, slope = 0.026±0.005 | $R^2 = 0.28$, **$p = 1 \times 10^{-4}$**, slope = 0.011±0.003 |
| School administration | $R^2 = 0.23$, **$p = 6 \times 10^{-4}$**, slope = 0.030±0.008 | $R^2 = 0.20$, p = 0.001 |
| General administration | $R^2 = 0.03$, p = 0.2 | $R^2 = 0.00$, p = 1.0 |
| "Other" support services | $R^2 = 0.17$, p = 0.003 | $R^2 = 0.20$, p = 0.001 |

is more of a descriptor of the state-to-state variability in gender diversity, as opposed to an explanation for why this variability exists, it is consistent with recent reports that students' perception of the gender bias in certain fields is a substantial contributor to selection of college

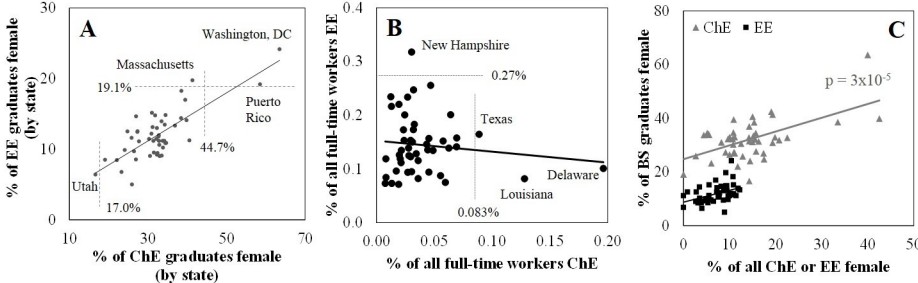

**Fig 4. State-by-state employment data.** (A) Significant association, by state, of gender diversity among new BS graduates in ChE and EE. Dashed lines indicate Tukey's fences, and outliers are labeled. On average, the pool of new ChE graduates is 2.9±0.3-fold more diverse than the pool of new EE graduates. B) The relative abundance of full-time chemical engineers, regardless of gender, is not significantly associated with the relative abundance of full-time electrical engineers. Dashed lines indicate Tukey's fences. Employment data for Puerto Rico was not available. C) Gender diversity among new BS graduates is significantly associated with the gender diversity of the existing workforce for ChE but not EE. Employment data for Puerto Rico was not available.

major [20], with the degree of perceived gender bias towards a particular field of employment being influenced by the gender composition of the workforce in that state.

## Regional variation: Funding of K-12 education

BS graduation data binned by state was also compared to K-12 educational spending per state, averaged over all years between the 1996–97 to 2011–2012 fiscal years (Fig 5, Table 3, S3 Table). On average, an individual state spent $8,472 per K-12 pupil during this time range, with this value ranging from $5,163 to $13,863. This spending can be parsed into three broad levels: instruction; support services; and "other" (Fig 5). For both ChE and EE, state funding of support services was significantly associated with gender diversity among ChE and EE BS graduates, while total spending, instruction spending and "other" spending did not meet the criterion of statistical significance (Table 3). These results indicate that not only is K-12 funding significantly associated with gender diversity at the BS level, but also that certain types of K-12 funding have more impact than others.

Instruction spending mainly includes salaries, wages and benefits and accounts for an average of 60% of the per-pupil spending (Fig 5). Support service spending accounts for an average of 35% of per-pupil spending and can be further parsed into: general administration; school administration; pupil support services; instructional staff support; and "other" (including maintenance). Among these various types of support services, only "instructional staff support services" was significantly associated with gender diversity among BS graduates for both ChE and EE (Table 3).

Instructional staff support, which showed a significant association with gender diversity, includes curriculum development, instructional staff training, and instruction services involving media, library, audiovisual and computers. School administration spending is associated with the office of principal services. Pupil support services include: attendance record-keeping; counseling; social work; medical, dental, nursing, psychological and speech services; student accounting; student appraisal; record maintenance; and placement services. General administration spending is associated with the board of education and office of the superintendent.

These results suggest that financial support of aspects of K-12 education outside of the traditional framework of "instruction" may have a long-reaching impact on the composition of the future workforce of that state.

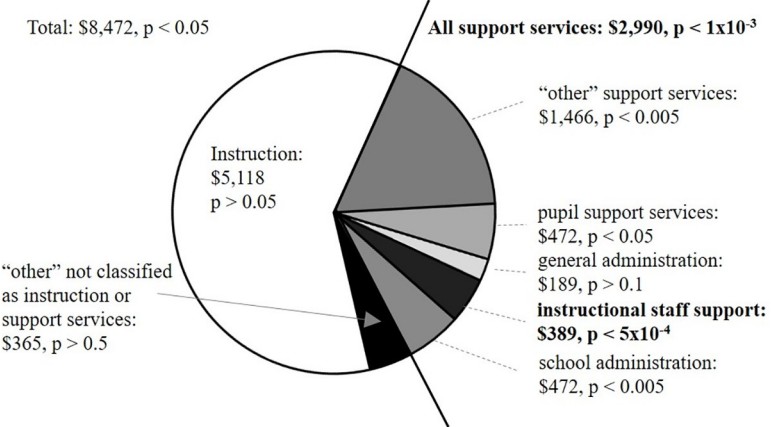

**Fig 5. State-wide diversity among BS graduates is significantly associated with K-12 expenditures toward support services.** State-specific spending data is presented as an average value, 1996–2012. P values indicate the values for both ChE and EE BS graduate gender diversity data, with bolded values indicating satisfaction of the $p < 0.001$ criterion. Numerical P values are provided for each field in Table 3.

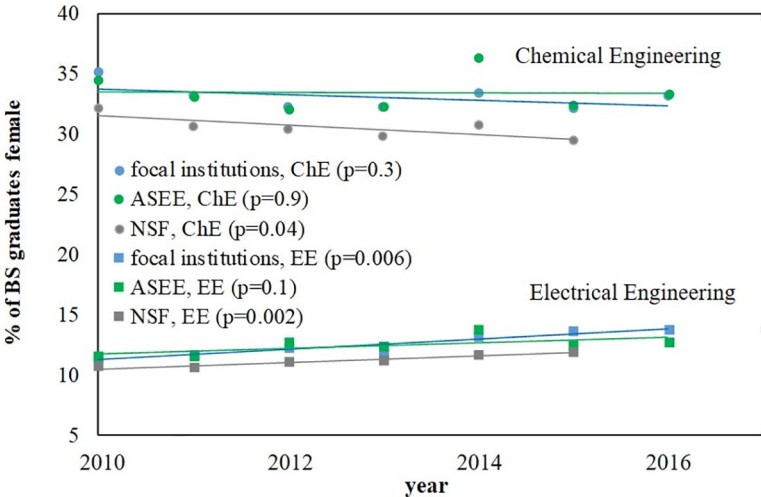

**Fig 6. Gender diversity among ChE and EE BS graduates over time.** Nationwide trends in ChE and EE BS gender diversity as tracked according to the 95 focal institutions used here, ASEE and NSF. 2016 NSF data was not available at the time of the preparation of this manuscript.

## Looking forward

The analyses described thus far all consider a static snapshot of the total BS graduates between 2010 and 2016. However, trends in gender diversity over time are also of interest (Fig 6). Among the 95 focal institutions, ASEE-reported data, and NSF-reported data, the gender diversity among ChE and EE BS graduates cannot be concluded to be either increasing or decreasing using the statistical criterion of $p < 0.001$. Additionally, no individual institutions were identified where the gender diversity among recent BS graduates in either field could be concluded to be significantly increasing or decreasing.

## Discussion

Here, we have assessed variation in gender diversity among recent BS graduates in Chemical Engineering and Electrical Engineering, two subfields that span the range of gender diversity in Engineering, the least gender-diverse STEM field. The most surprising and actionable finding of this work is that state support of K-12 education at the level of instructional staff support, which includes curriculum development, instructional staff training, is significantly associated with increased gender diversity in both ChE and EE (Fig 5). It is not clear if this trend extends to other types of STEM and to other under-represented groups. It has previously been shown that increased spending outside of standard instruction reduced the risk of work-related injuries against educators [33], but to the best of our knowledge this is the first report associating support service spending at the K-12 level to gender diversity among college graduates. It should be noted that any relocation between states during the course of K-12 and university-level education, or after completion of a BS degree would not be accounted for in the analysis presented here.

Institution type, particularly the distinction between private and public institutions and HBCUs and non-HBCUs was also found to significantly impact gender diversity in both fields (Fig 2, Table 2). In contrast to previous reports [29], we did not observe a significant association between tenure-track faculty gender diversity and recent BS graduate diversity for either ChE or EE (Fig 2C). However, our analysis was restricted to reported faculty composition and not actual teaching loads and some studies have also reported a lack of association, or even a

negative association, between female instructors and persistence of female students within STEM curriculum [31, 34]. A lack of significant impact by faculty gender is consistent with reports that disparate interest in engineering begins as early as middle school [10], especially when considering that K-12 educational support was found to have a significant impact.

The perception of gender discrimination has been demonstrated to be a substantial driver of career selection for female students [20], and deviation from stereotyped gender roles is differentially costly for females relative to males [8]. The strength of the gender stereotype for ChE and EE, and the perception of gender discrimination in these fields, may differ regionally. This type of regional variability is consistent with our finding that states with a larger percentage of working chemical engineers who are female tend to have higher gender diversity among recent BS graduates in these fields. These trends are also consistent with reports that interaction with a female STEM expert increased the commitment of female students to pursuing a STEM career [35] and that pairing of first-year engineering students with other female engineering students increases retention [16, 17]. This type of deliberate exposure of female students, both at the K-12 level and at the undergraduate level, to female engineers may be especially impactful in regions with relatively few practicing female engineers in the workforce. Regional variability could also possibly be mitigated by increased representation of female engineers in the media and popular culture. For example, engineers portrayed in popular television and film 2007–2017 were five times more likely to be male than female [36].

We observed several differences between EE and ChE. Across institution types and census regions, most EE departments produce more BS graduates than ChE departments at the same institution (Fig 1A), but with lower gender diversity among both recent BS graduates (Fig 1B) and tenure-track faculty (Fig 2B). EE departments grapple with the existence of Computer Engineering as a separate department or combined with EE. The existence of a separate degree in Computer Engineering is also associated with decreased gender diversity (Fig 3C). ChE departments are perceived as having a similar relationship with the various types of bioengineering, though no significant associations were identified. Though not investigated here, the, on average, 2-fold difference in gender diversity in these two fields may be influenced by the historical roots of these fields. Specifically, electric engineering largely grew out of Physics, while chemical engineering grew out of Chemistry. NSF reporting indicates that in the years 2000–2015 49.6% of the BS degrees earned in Chemistry were earned by females, while only 20.4% of the BS degrees earned in Physics were earned by females [4].

## Supporting information

**S1 Table. Institutional data.**
(PDF)

**S2 Table. Data lumped by census region and institution type.**
(PDF)

**S3 Table. State-specific data.**
(PDF)

## Acknowledgments

We thank David Knight for helpful discussion during preparation of this manuscript.

## Author Contributions

**Conceptualization:** Laura R. Jarboe.

**Data curation:** Laura R. Jarboe.

**Formal analysis:** Laura R. Jarboe.

**Investigation:** Laura R. Jarboe.

**Methodology:** Laura R. Jarboe.

**Project administration:** Laura R. Jarboe.

**Validation:** Laura R. Jarboe.

**Visualization:** Laura R. Jarboe.

**Writing – original draft:** Laura R. Jarboe.

**Writing – review & editing:** Laura R. Jarboe.

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
