## [Decision Letter · Decision Letter 0]

17 Sep 2019

PONE-D-19-21208

Regional, institutional, and departmental factors associated with gender diversity among BS-level Chemical and Electrical Engineering graduates

PLOS ONE

Dear Dr. Jarboe,

I sincerely apologize for how long this review took.  We are excited to have it at PLOS One - please make the suggested minor revisions. 

Thank you for submitting your manuscript to PLOS ONE. After careful consideration, we feel that it has significant merit but does not fully meet PLOS ONE’s publication criteria as it currently stands. Therefore, we invite you to submit a revised version of the manuscript that addresses the few and minor points raised during the review process.

Both the reviewers and myself thought that this topic was intriguing and a necessary study.  The manuscript was well written and if you can make the minor revisions (suggested by reviewers) that will allow for me to provide full acceptance of this article into PLOS One and then move the manuscript quickly through to publication.

We would appreciate receiving your revised manuscript by Nov 01 2019 11:59PM. To enhance the reproducibility of your results, we recommend that if applicable you deposit your laboratory protocols in protocols.io, where a protocol can be assigned its own identifier (DOI) such that it can be cited independently in the future. For instructions see: http://journals.plos.org/plosone/s/submission-guidelines#loc-laboratory-protocols

We look forward to receiving your revised manuscript.

Kind regards,

Kristopher V. Waynant, Ph.D.

Academic Editor

PLOS ONE

Journal Requirements:

Reviewers' comments:

Reviewer's Responses to Questions

**Comments to the Author**

1. Is the manuscript technically sound, and do the data support the conclusions?

Reviewer #1: Yes

Reviewer #2: Yes

2. Has the statistical analysis been performed appropriately and rigorously? 

Reviewer #1: Yes

Reviewer #2: Yes

3. Have the authors made all data underlying the findings in their manuscript fully available?

Reviewer #1: Yes

Reviewer #2: Yes

4. Is the manuscript presented in an intelligible fashion and written in standard English?

Reviewer #1: Yes

Reviewer #2: Yes

5. Review Comments to the Author

Reviewer #1: The present manuscript looks to identify various regional, institutional, and departmental factors correlated with gender diversity in BS-level Chemical and Electrical Engineering. Overall, the manuscript is well written and the data appears to support the conclusions.

Comments:

1) I could not find figure 1A. Please include it in the revised manuscript.

2) the manuscript states that "On average, ChE departments have 45+/-4% higher gender diversity among BS graduates the EE departments." Based on the figure, it look like ChE departments have nearly double the gender diversity of EE departments. Please clarify this and any similar calculations in the manuscript.

Reviewer #2: • What are the main claims of the paper and how significant are they for the discipline?

The % of female BS Chemical Engineering (ChE) graduates is an exemplar of high gender diversity among engineering disciplines whereas BS Electrical Engineering (EE) graduates exhibit a very low female fraction, both regardless of tenure-track female faculty representation (viz., not correlated). State industry female representation has a positive association with ChE but none with EE. K12 support service funding associate positively with gender diversity in both fields. Gender trends appear stable in both fields.

While these data are of great importance to many in STEM fields, especially those in engineering and ChE and EE educators in particular, the author has not discussed the significance (beyond the statistical, that is).

• Are the claims properly placed in the context of the previous literature? Have the authors treated the literature fairly? [Yes.]

• Do the data and analyses fully support the claims? If not, what other evidence is required?

The claims are very modest and entirely statistical, in my assessment.

• PLOS ONE encourages authors to publish detailed protocols and algorithms as supporting information online. Do any particular methods used in the manuscript warrant such treatment? [Not beyond what appears in the manuscript.] If a protocol is already provided, for example for a randomized controlled trial, are there any important deviations from it? [No.] If so, have the authors explained adequately why the deviations occurred? [N/A]

• If the paper is considered unsuitable for publication in its present form, does the study itself show sufficient potential that the authors should be encouraged to resubmit a revised version? [Yes.]

• Are original data deposited in appropriate repositories and accession/version numbers provided for genes, proteins, mutants, diseases, etc.? [N/A]

• Are details of the methodology sufficient to allow the experiments to be reproduced? [N/A]

• Is the manuscript well organized and written clearly enough to be accessible to non-specialists? [Yes.]

 

Suggestions for Improvements

Check subject-verb agreements with criteria and data (both plural), and use of criteria where “criterion” may be the intended word for use.

P14 line 276 – “...associated significantly associated (sic) with ...”

P20 Discussion – some confusion between “instructional staff support services” mentioned in Results (p19, lines 366-69) to correlate with gender diversity at the BS level, and then the use of the term non-instructional “support services” in line 399 that seems to conflict with K-12. This could be a simple oddity of how each institutional level categorizes/labels their spending, or it may be a much deeper point to consider, as it seems to be phenomenological in functional contradiction to what actually makes a difference at different levels of instruction, how they are executed, and/or some other causal connection(s).

Recognizing that this is not precisely within the paper’s scope, a quick reference and correlation of EE to Physics and ChE to Chemistry at least as a parting comment in the Discussion might elucidate or at least point toward a similarity in a more robust and even causal/historical context than comparing EE with CompE to ChE with Bioengineering/Bio-related degrees. Many ChE programs and industry positions grew historically from chemistry, whereas many EE programs arose historically from Physics. The gender diversity of these two additional science disciplines should be reported/referenced at least in passing—perhaps that is a lead-in to a follow-up, causation study made as a concluding remark herein for the author’s next project.

6. PLOS authors have the option to publish the peer review history of their article (what does this mean?). If published, this will include your full peer review and any attached files.

Reviewer #1: No

Reviewer #2: No

---

## [Author Response · Author response to Decision Letter 0]

22 Sep 2019

We thank the reviewers for their thorough review and helpful suggestions. A point-by-point response is provided below.

Reviewer #1

Comment: I could not find figure 1A. Please include it in the revised manuscript.

Response: It seems that Figures 1 and 2 were placed in incorrect order in the manuscript. This has been corrected.

Comment: the manuscript states that "On average, ChE departments have 45+/-4% higher gender diversity among BS graduates the EE departments." Based on the figure, it look like ChE departments have nearly double the gender diversity of EE departments. Please clarify this and any similar calculations in the manuscript.

Response: Thank you very much for catching this discrepancy. This had been updated, as has a similar error in the “regional variation: employment trends” section.

Reviewer #2: 

Comment: The % of female BS Chemical Engineering (ChE) graduates is an exemplar of high gender diversity among engineering disciplines whereas BS Electrical Engineering (EE) graduates exhibit a very low female fraction, both regardless of tenure-track female faculty representation (viz., not correlated). State industry female representation has a positive association with ChE but none with EE. K12 support service funding associate positively with gender diversity in both fields. Gender trends appear stable in both fields.

While these data are of great importance to many in STEM fields, especially those in engineering and ChE and EE educators in particular, the author has not discussed the significance (beyond the statistical, that is).

Comment: The claims are very modest and entirely statistical, in my assessment.

Response: The reviewer is correct that I have been cautious in interpretation of these results. I hope that my presentation of these results will spur further analysis by those with expertise in this field. I have added a few more sentences to the discussion in response to this comment.

Comment:

- Check subject-verb agreements with criteria and data (both plural), and use of criteria where “criterion” may be the intended word for use.

Response: Done, thank you.

- P14 line 276 – “...associated significantly associated (sic) with ...”

Response: Done, thank you.

- P20 Discussion – some confusion between “instructional staff support services” mentioned in Results (p19, lines 366-69) to correlate with gender diversity at the BS level, and then the use of the term non-instructional “support services” in line 399 that seems to conflict with K-12. This could be a simple oddity of how each institutional level categorizes/labels their spending, or it may be a much deeper point to consider, as it seems to be phenomenological in functional contradiction to what actually makes a difference at different levels of instruction, how they are executed, and/or some other causal connection(s).

Response: Yes! I agree that the descriptors used by the Educational Finance Branch are very confusing. But, they are what they are. I have edited the text, Figure 5 and Table 3 in an effort to increase the clarity of how these various spending types are presented while staying true to the language used in the source documents.

- Recognizing that this is not precisely within the paper’s scope, a quick reference and correlation of EE to Physics and ChE to Chemistry at least as a parting comment in the Discussion might elucidate or at least point toward a similarity in a more robust and even causal/historical context than comparing EE with CompE to ChE with Bioengineering/Bio-related degrees. Many ChE programs and industry positions grew historically from chemistry, whereas many EE programs arose historically from Physics. The gender diversity of these two additional science disciplines should be reported/referenced at least in passing—perhaps that is a lead-in to a follow-up, causation study made as a concluding remark herein for the author’s next project.

Response: Thank you for this suggestion. This information has been added to the final paragraph of the discussion section.

---

## [Editor Report · Decision Letter 1]

25 Sep 2019

Regional, institutional, and departmental factors associated with gender diversity among BS-level Chemical and Electrical Engineering graduates

PONE-D-19-21208R1

Dear Dr. Jarboe,

We are pleased to inform you that your manuscript has been judged scientifically suitable for publication and will be formally accepted for publication once it complies with all outstanding technical requirements.  Thank you for looking through the reviews and making the appropriate changes.

With kind regards,

Kristopher V. Waynant, Ph.D.

Academic Editor

PLOS ONE

Additional Editor Comments (optional):  Thank you for your report on a very serious issue.
---

## [Editor Report · Acceptance letter]

30 Sep 2019

PONE-D-19-21208R1 

Regional, institutional, and departmental factors associated with gender diversity among BS-level Chemical and Electrical Engineering graduates 

Dear Dr. Jarboe:

I am pleased to inform you that your manuscript has been deemed suitable for publication in PLOS ONE. Congratulations! Your manuscript is now with our production department. 

With kind regards,

on behalf of

Dr. Kristopher V. Waynant 

Academic Editor

PLOS ONE